# Magnesium and Drugs

**DOI:** 10.3390/ijms20092094

**Published:** 2019-04-28

**Authors:** Uwe Gröber

**Affiliations:** Academy of Micronutrient Medicine, Zweigertstr. 55, 45130 Essen, Germany; uwegroeber@gmx.net; Tel.: +49-201-874-2984

**Keywords:** Magnesium, drugs, drug-induced magnesium deficiency, proton-pump inhibitors, diuretics, TRPM6, SLC41A1

## Abstract

Several drugs including diuretics and proton-pump inhibitors can cause magnesium loss and hypomagnesemia. Magnesium and drugs use the same transport and metabolism pathways in the body for their intestinal absorption, metabolism, and elimination. This means that when one or more drug is taken, there is always a potential risk of interaction with the magnesium status. Consequently the action of a drug may be adversely affected by magnesium (e.g., magnesium, calcium, and zinc can interfere with the gastrointestinal absorption of tetracycline antibiotics) and simultaneously the physiological function of minerals such as magnesium may be impaired by a drug (e.g., diuretics induce renal magnesium loss). Given the ever-increasing number of drugs on the market and the frequency with which they are used, greater attention must be paid in daily medical and pharmaceutical practice focused in particular on the adverse effects of drug therapy on magnesium status in order to minimize the potential risk to the health of patients.

## 1. Introduction

Thanks to modern health care and the improvement of life quality, the average life expectancy of Europeans has almost doubled over the past 100 years. Consequently, in the European Union the demographic old-age dependency ratio will continue to rise significantly over the coming decades due to a large increase in the population above 65 years old. From being about 25% in 2010, it has risen to 29.6% in 2016 and is projected to rise further, in particular up to 2050, and eventually reach 51.2% in 2070 [1]. The increase in the mean age is associated with an increasing number of multimorbid patients, who suffer from nutrition-associated diseases and usually depend on complex pharmacotherapy [2]. As example, a population study in Scotland found an overall prevalence of multimorbidity of 23.2% [3]. A recent cross-sectional analysis in Germany with more than 10,000 participants aged 50 years and older revealed that even more than 95% of the patients with osteoporosis had at least one coexisting disease [4]. The prevalence of multimorbidity in a population increases with age and leads inevitably to polypharmacotherapy. Polypharmacotherapy is a major concern in the elderly [5,6]. 40% of institutionalized patients take more than nine drugs daily [7]. Each additional medication, however, increases the risk of adverse drug reactions [8,9]. In the year 1998 a meta-analysis of 39 prospective studies from US hospitals has shown that the overall incidence of serious adverse drug reactions was 6.7% (95% CI: 5.2–8.2%) and of fatal adverse drug reactions was 0.32% (95% CI: 0.23–0.41%) of hospitalized patients. In this analysis adverse drug reactions were the fourth and sixth leading cause of death. Although the results of this analysis should not be overrated given the heterogeneity of the conducted studies, it becomes clear that adverse drug reactions are of high clinical relevance and can take on critical dimensions [10,11,12,13,14,15]. 

## 2. Magnesium and Drugs

Drugs and micronutrients use the same transport and metabolism pathways in the body for their intestinal absorption, metabolism, and elimination. This means that when one or more drugs are taken, there is always a potential risk of interactions with the nutrient status. Consequently the action of a drug may be adversely affected by a micronutrient (e.g., magnesium, calcium and zinc can interfere with the gastrointestinal absorption of tetracycline antibiotics) and simultaneously the physiological function of a mineral such as magnesium may be impaired by a drug (e.g., thiazide diuretics induce renal magnesium loss) [14,15]. Disruption of micronutrient status can result in serious metabolic dysfunctions, as there is hardly a single physiological process in the body that is not mediated by one or other of these biocatalysts. Given the ever-increasing number of drugs on the market and the frequency with which they are used, greater attention must be paid in daily medical and pharmaceutical practice focused in particular on the adverse effects of drug therapy on the micronutrient status (Figure 1) in order to minimize the potential risk to the health of patients. This review aims to sensitize physicians and pharmacists on important drug magnesium interactions with selected examples of widely prescribed drugs that can precipitate magnesium deficiency.

## 3. Influencing Factors

The magnesium concentration in blood is regulated by a dynamic balance and interplay between three organs: the intestine (facilitating magnesium uptake), the bone (the magnesium storage system: availability of magnesium to maintain constant serum levels) and the kidneys (renal transport and excretion). The interference of drugs with the magnesium homeostasis can be influenced by many different factors, such as age, pH value, life style (e.g., diet alcohol) and/or genetic defects of magnesium-transporting proteins. 

### 3.1. Age

Children are not small adults. Compared to adults the drug and/or mineral disposition and metabolism in children is different. Generally known, age-dependent changes in body function alter the pharmacokinetic parameters that determine each compound’s duration of action, extent of drug receptor interaction and the drug’s rates of absorption, distribution, metabolism, and excretion. 

Important factors for *absorption* of drugs and micronutrients are the gastric emptying, gastric acid production, and intestinal transit time. Gastric acid secretion approaches the lower limit of adult values by 3 months of age. Both gastric emptying time and small intestine peristalsis tend to be slow the later part of the first year of life. Key factors in drug distribution are membrane permeability, plasma protein, endogenous substances in plasma, total body and extracellular water, fat content, and regional blood flow. Newborns have decreased plasma albumin and total plasma protein concentrations. Their albumin shows a decreased drug-binding affinity. This results in increased plasma level of free drugs and the potential for toxicity. Great differences have also been found by hepatic drug metabolism (e.g., phase I enzyme and phase II enzyme reactions). Although the cytochrome P 450 system is fully developed at birth, it functions more slowly than in adults. Infants and children have greater capacity to carry out sulfate conjugation than do adults. For example, acetaminophen is excreted predominantly as sulfate conjugate in children compared as glucuronide conjugate in adults. 

Finally, there are differences in renal excretion, glomerular filtration, and tubular secretion. For example, preterm infants have glomerular filtration rates approximately one-tenth of a term newborn. Because of limitations of tubular reabsorption, they have increased urinary loss of filtered substances. Newborns require less frequent dosing interval for many drugs. For example, aminoglycosides are administered every 8 hours in older children, every 12 hours in newborns, and every 24 hours in premature infants. Paradoxically drugs such as phenobarbital, which have a sedating action on adults, may produce hyperactivity in children [16,17,18].

Despite recent advances in this area, knowledge of the action and disposition of drugs and/or minerals in children is limited. This lack of information has made drug therapy for children difficult and dangerous. Drug administration must be tailored to meet the unique needs of children at their varied stages of development.

### 3.2. pH Value

Intestinal magnesium absorption occurs via a passive, nonsaturable paracellular pathway and an active, saturable transcellular pathway. Paracellular magnesium absorption is responsible for 80–90% of intestinal magnesium uptake. A minor, yet important, regulatory fraction of magnesium is *active* transported via the transient receptor potential channel melastatin member 6 (TRPM6) and 7 (TRPM7). TRPM6 expression is mainly detected in the distal small intestine and colon, whereas TRPM7 is ubiquitously expressed. As the pH value of the gastrointestinal tract is part of essential physiologic processes including digestion and nutrient (e.g., magnesium) absorption, drugs such as proton-pump inhibitors that suppress gastric acid can interfere with both mechanisms the passive and active magnesium absorption [19,20,21,22]. It has been shown that as the pH value gradually increases, the solubility of different magnesium salts (organic and inorganic) decreases from 85% in the proximal intestine to 50% in the distal intestine. The proton-pump inhibitor omeprazole suppresses passive magnesium absorption by causing luminal acidity to rise above the range (pH 5.5–6.5) in which claudin 7 and 12 expression is optimized [23,24,25]. In the elderly the prevalence of atrophic gastritis and hypochlorhydria in association with the frequency of Helicobacter pylori infection is high. Atrophic gastritis leads to failures in the secretion of hydrochloric acid and intrinsic factor.

In acid-free and atrophic stomach, due to the impairment in the secretion of hydrochloric acid and/or intrinsic factor, absorption of micronutrients such as magnesium and vitamin B12 is impaired [25,26]. Even the diet can contribute to low-grade metabolic acidosis and increase blood pH through the ingestion of dietary constituents of non-volatile acids and bases. As blood pH increases the magnesium concentration decreases, indicating a stronger binding of magnesium with proteins in the more alkaline environment. The increased dietary acid load leads to small changes in the acid-base balance (increase in H^+^ and reduction in HCO_3_^−^) and induces a low-grade metabolic acidosis. Nutrients that release H^+^ precursors in the bloodstream are mainly protein components (e.g., sulfur amino acids such as methionine, cysteine). Although these changes in the acid-base balance are small, it has been shown that a diet-induced slight decrease in blood pH can have a significant impact on metabolism (e.g., bone) and mineral excretion [25,26].

The kidneys are the key player in magnesium homeostasis that filter approximately 2400 mg magnesium daily, and up to 70% of the magnesium filtered can be excreted in the urine. This wide range depends on ever changing variables such as dietary intake, existing magnesium status, mobilization from bone and muscle, and the influence of a variety of hormones (e.g., parathyroid hormone, calcitonin) and drugs (e.g., diuretics) [25,26,27].

### 3.3. Diet, Lifestyle

Dietary habits can additively aggravate the negative impact of the drugs on magnesium status. In the US and the UK, the magnesium content of vegetables and fruit have lost in the past 100 years large amounts of minerals and other micronutrients. It is estimated that vegetables (e.g. cabbage, lettuce, spinach) have dropped magnesium levels by around 80–90%. Modern dietary practices such as vegetable cooking and grain bleaching can cause a loss of up to 80% of magnesium content. Soft drinks, which contain high phosphoric acid, along with a low protein diet and foods containing phytates and oxalates (e.g., rice, nuts), all contribute to magnesium deficiency due to their ability to bind magnesium to produce insoluble precipitates that decrease availability and absorption of magnesium. Additionally, the ingestion of alcohol and caffeine increase the renal excretion of magnesium causing an increase in the body’s demand) [25,26,27,28].

### 3.4. Magnesium Transporters

Regarding the interaction of drugs with the magnesium homeostasis the interference of commonly prescribed drugs (e.g., antidepressants, omeprazole, insulin mimetic drugs) with different magnesium-transporting proteins is gaining increased importance. A plethora of biochemical processes requires a tight regulation of intracellular magnesium homeostasis. One of the main factors influencing the levels of cytosolic free magnesium is the activity of transport systems in the plasma membrane and mitochondria and the concentration of nucleotides, such as adenosintriphosphate (ATP). ATP, for example, can bind magnesium with a high dissociation constant. The intracellular magnesium concentration is regulated by the modulation of cellular uptake and/or efflux or by intracellular storage. Thus, several transporters or channels have been characterized as mediating the uptake of magnesium or extrusion across the cytoplasmic or mitochondrial membrane. In the last 10 years genetic screenings on human diseases have resulted in the identification of numerous magnesium-transporting proteins, such as the ubiquitous transient receptor potential melastatin type 7 (TRPM7) or the solute carrier family 41 member 1 (SLC41A1) and the tissue-specific transient receptor potential melastatin type 6 (TRPM6: kidney, colon) or the solute carrier family 43 (SLC41A3: mitochondria) [27,29,30,31,32,33,34,35].

For example, it has been recently shown that insulin can not only inhibit the Na^+^/Mg^2+^ exchanger function of SLC41A1 but also promote the early efflux of Mg^2+^ from mitochondria or other organelles, thereby supporting the maintenance of physiological Mg^2+^ concentration in the cytoplasm. A hyperinsulinemic condition is associated with an increased magnesium efflux. Thus, the intracellular compartmentalization of magnesium is influenced by insulin. This could explain the high prevalence of hypomagnesemia and/or magnesium deficiency in patients with diabetes. These data underline the importance of an adequate magnesium status in diabetic patients treated with insulin or insulin mimetic drugs and provide a molecular target for further studies of the prevention and treatment of diabetes associated sequelae [36,37,38].

Some antipsychotic and antidepressant drugs such as imipramine can also interfere with the Na^+^/Mg^2+^ exchanger SLC41A1. Imipramine for instance is an inhibitor of the SLC41A1 magnesium efflux [37,39,40]. Also, signalome modulating drugs can impact the performance of TRPM7. Under these drugs the magnesium status should be regularly controlled.

## 4. Drug-Induced Magnesium Deficiency

Hypomagnesemia is frequently associated with other electrolyte abnormalities such as hypokalemia and hypocalcemia. Conditions that may lead to hypomagnesemia include alcoholism, poorly controlled diabetes, malabsorption (e.g., Crohn’s disease, ulcerative colitis, coeliac disease, short bowel syndrome, Whipple’s disease), endocrine causes (e.g., aldosteronism, hyperparathyroidism, hyperthyroidism), renal disease (e.g., chronic renal failure, dialysis, Gitelman’s syndrome) and medication use. A variety of drugs including antibiotics, chemotherapeutic agents, diuretics and proton-pump inhibitors can cause magnesium loss and hypomagnesemia (see Table 1) [10,27,28,33,34,39,41,42].

### 4.1. Proton-Pump Inhibitors (PPIs)

First introduced in 1989, proton-pump inhibitors (PPIs) are among the most widely used medications worldwide, both in the ambulatory and inpatient clinical settings, with a high inappropriate prescription rates, exceeding partially 50% in the elderly [42,43,44,45]. PPIs block the gastric H^+^/K^+^-ATPase, inhibiting gastric acid secretion. This effect enables healing of peptic ulcers, gastroesophageal reflux disease (GERD), Barrett’s esophagus, and Zollinger-Ellison syndrome, as well as the eradication of Helicobacter pylori as part of combination regimens. However, by increasing the intragastric pH, PPIs can impair the absorption and use of micronutrients such as magnesium, calcium, iron, vitamin C, and vitamin B12. This can induce clinical problems related to the deficiencies of these micronutrients (e.g., hypomagnesemia, anemia, vitamin B_12_ deficiency) [45].

#### 4.1.1. PPIs and Magnesium

Magnesium is primarily found within the cell where it acts as a counter ion for the energy-rich ATP and nuclear acids. Magnesium is a cofactor in more than 600 pacemaker enzyme systems, encompassing approximately 80% of all known metabolic functions, that regulate elementary biochemical reactions in the body, including protein synthesis, muscle and nerve transmission, neuromuscular conduction, blood glucose control, and blood pressure regulation. Some magnesium-dependent enzymes are Na^+^/K^+^-ATPase, hexokinase, creatine kinase, protein kinase, and cyclases. Magnesium is also necessary for structural function of proteins, nucleic acids, or mitochondria. It is required for DNA and RNA synthesis, reproduction, and for both aerobic and anaerobic energy production—oxidative phosphorylation and glycolysis—either indirectly as a part of magnesium-ATP complex, or directly as an enzyme activator.

Magnesium also plays a key role in the active transport of calcium and potassium ions across cell membranes, a process that is important to nerve impulse conduction, muscle contraction, vasomotor tone, and normal heart rhythm. Hypomagnesemia is frequently linked with hypokalemia owing to disturbances in renal K+ secretion in the connecting tubule and collecting duct. Magnesium is a natural calcium antagonist—the block of *N*-methyl-d-aspartate (NMDA) receptor channels by external magnesium is believed to be of great physiological importance. Moreover, it contributes to the structural development of bone and is required for the adenosine triphosphate-dependent synthesis of the most important intracellular antioxidant glutathione. Magnesium absorption and excretion is influenced by different hormones. It has been shown that 1,25-dihydroxyvitamin D [1,25(OH)_2_D] can stimulate intestinal magnesium absorption. On the other hand, magnesium is a cofactor that is required for the binding of vitamin D to its transport protein, vitamin D binding protein (VDBP). Moreover, conversion of vitamin D by hepatic 25-hydroxlation and renal 1α-hydroxylation into the active, hormonal form 1,25(OH)_2_D is magnesium-dependent. Magnesium deficiency, which leads to reduced 1,25(OH)_2_D and impaired parathyroid hormone response, has been implicated in ”magnesium-dependent vitamin-D-resistant rickets”. Magnesium supplementation substantially reversed the resistance to vitamin D treatment [23,27,28,45,46,47,48,49].

Of special importance is parathyroid hormone (PTH). Absorption of both magnesium and calcium appears to be inter-related, with concomitant deficiencies of both ions well described. A common link is that of PTH, secretion of which is enhanced in hypocalcemia. Hypomagnesemia impairs hypercalcemic-induced PTH release, which is corrected within in minutes after infusion of magnesium. The rapidity of correction of PTH concentrations suggests that the mechanism of action of magnesium is enhanced release of PTH. PTH release enhances magnesium reabsorption in the kidney, absorption in the gut and release from the bone [28,46,47,48,49].

Long-term use of PPIs has been associated in some cases with hypomagnesemia, hypocalcemia, and hypoparathyroidism. Since the year 2006 there have been more than 40 reported cases of PPI-induced hypomagnesemia (<0.76 mmol/L). Thus, in 2011, the US Food and Drug Administration (FDA) published a safety announcement, including hypomagnesemia as a long-term side-effect of PPI based on accumulating evidence. A severe magnesium deficiency with PPI use can occur in rare cases and accounts for less than 1% of all PPI-induced side effects voluntarily reported to the FDA. Others reported hypomagnesemia in 13% of PPI users. Nevertheless, patients on long-term treatment with PPIs should be monitored for magnesium deficiency, especially those with diabetes and cardiovascular disease (e.g., arrhythmias, heart failure, hypertension), because the probability and the risk of disease-associated complications is increased [46,50,51]. The mechanisms by which PPIs can interfere with intestinal magnesium absorption is still being investigated. PPIs such as omeprazole may decrease intestinal magnesium absorption by interfering with both passive (paracellular pores) absorption and active (transient receptor potential melastatin protein channels, TRPM6 and TRPM7) absorption. Common single nucleotide polymorphisms in the gene TRPM6 have been identified and are discussed to be responsible for the risk of hypomagnesemia in PPI treated patients [50,51,52,53].

Magnesium insufficiency and hypomagnesemia (<0.76 mmol/L) are a relatively common in clinical practice. There are often unrecognized because of the fact that magnesium status is rarely controlled since only a few clinicians are aware of the many clinical states in which magnesium deficiency can manifest. Despite normal serum magnesium levels, a magnesium deficiency can be present [51,54,55]. In older publications the prevalence of marginal magnesium deficiency in developed countries is estimated to be 15 to 20%. Recent population-based cross-sectional studies and clinical trials indicate that some 10 to 30% of a given population, considered healthy, have a subclinical magnesium deficiency based on serum magnesium levels <0.80 mmol/L [54,55,56,57,58]. In postmenopausal women with osteoporosis magnesium deficiency has been found in 84% diagnosed by low magnesium trabecular bone content and Thoren’s magnesium load test. In an ageing population, the number of patients treated with diuretics is increasing, as is the significance of diuretic therapy-associated side effects. In the elderly the prevalence of magnesium and potassium deficiencies is about 20% [59]. The concentration of intracellular ionized magnesium is physiologically relevant. Thus, ionized magnesium in erythrocytes is one of the best laboratory parameters to judge magnesium deficiency. Among critically ill postoperative patients, 36.5% were found to have magnesium deficiency based on ionized magnesium levels in red blood cells [39]. Based on a reference range for serum magnesium <0.76 mmol Mg/L the frequency of hypomagnesemia was evaluated in an unselected population group of about 16,000 individuals of Germany. Hypomagnesemia was present in about 14.5% of all individuals with generally higher frequencies in females and outpatients. In the elderly, especially in old ladies, the prevalence was highest and concerned about 30% of this subgroup. In general, in this study suboptimal magnesium levels were observed in 33.7% of the population [59,60,61].

Hypomagnesemia can cause serious neuromuscular and cardiovascular problems and is often accompanied by hypovitaminosis D, hypocalcemia, and hypokalemia. Early signs of magnesium deficiency are nonspecific and include loss of appetite, lethargy, nausea, vomiting, fatigue, muscle cramps, weakness, and lethargy. More pronounced magnesium deficiency presents with symptoms of increased neuromuscular irritability such as tremor of the extremities, tetany, generalized seizures, or convulsions. Hypomagnesemia can cause hypokalemia with electrocardiogram changes and cardiac arrhythmias including atrial and ventricular tachycardia, prolonged QT interval, and torsades de pointes. Carpopedal spasms are often associated with hypoparathyroidism, hypovitaminosis D, and/or hypocalcemia. Additionally, the neuromuscular symptoms of hypomagnesemia (e.g., convulsions, muscle weakness, tetany) can also be related to the co-existent hypovitaminosis D [25(OH)D: <20 ng/mL] and/or hypocalcemia [28,49,61,62,63].

#### 4.1.2. Diagnostic of Magnesium Deficiency

Clinical diagnosis of magnesium deficiency is not simple, as symptoms associated with magnesium deficiency are unspecific, and generally confounded by low consumption of other nutrients. The most common and valuable test in clinical medicine for the rapid assessment of changes in magnesium status is the serum magnesium concentration, even though serum levels have little correlation with total body magnesium levels or concentrations in specific tissues. In healthy individuals, magnesium serum concentration is closely maintained within the physiological range. The normal reference range for the magnesium in blood serum is 0.76–1.05 mmol/L. A serum magnesium <0.82 mmol/L (2.0 mg/dL) with a 24-hour urinary magnesium excretion of 40–80 mg per day is highly suggestive of magnesium deficiency. According to many magnesium researchers the appropriate lower reference limit of the serum magnesium concentration should be 0.85 mmol/L, especially for patients with diabetes [23,26,28,34]. Emerging evidence suggests that the serum magnesium/calcium quotient (0.4 is optimal, 0.36–0.28 too low) is a more practical and sensitive indicator of magnesium status and/or turnover, than the serum magnesium level alone [23,26]. Intravenous or oral magnesium loading tests used in combination with magnesium excretion concentrations from a 24-h urine specimen may be more useful for detecting subclinical magnesium deficiency. It needs to be mentioned, however, that intravenous or oral magnesium loading tests require normal kidney or gastrointestinal functions to get an accurate reflection of magnesium status [23]. Furthermore, the ionized magnesium concentration has been shown to be accurate. The reference range for serum ionized magnesium concentration is 0.54–0.67 mmol/L. To comprehensively evaluate magnesium status, both laboratory tests and the clinical assessment of magnesium deficit symptoms might be required [23,26,28,34].

#### 4.1.3. Recommendations for Clinical Practice

Patients on long-term treatment with PPIs should be monitored for magnesium deficiency, particularly those with additive risk factors, such as therapy with use of diuretics, diabetes, cardiovascular diseases (e.g., hypertension, arrhythmias), inadequate dietary intake, secondary aldosteronism, and kidney dysfunction. Many nutritional experts feel the ideal intake for magnesium should be based on the body weight (e.g., 4–6 mg per kg/day). In the treatment of magnesium deficiency supplements with organic bound magnesium salts are recommended, such as magnesium citrate or gluconate. However, also mineral water with a high concentration of magnesium (>100 mg magnesium per litre) is a good source of magnesium that contributes to daily magnesium supply [28,64].

### 4.2. Thiazide Diuretics

Thiazide-type diuretics are the second most commonly prescribed class of antihypertensive medication, and thiazide-related diuretics have increased at a rate greater than that of antihypertensive medications as a whole. For more than 5 decades thiazide diuretics (TD), including thiazide-type (e.g., hydrochlorothiazide chlorothiazide) and thiazide-like diuretics (e.g., indapamide, chlorthalidone) have been used for the treatment of hypertension. The latest hypertension guidelines have underscored the importance of TD for all patients, but particularly for those with salt-sensitive and resistant hypertension [65]. Thiazide-type diuretics decrease efficaciously systolic and diastolic blood pressure and reduce at the same time cardiovascular morbidity and mortality associated with hypertension. A meta-analysis including 19 randomized controlled trials enrolling 112,113 patients showed that TD have an additional cardioprotective effect. During a mean follow up of 3.91-years, a 14% reduction in the risk of cardiac events (odds ratio (OR): 0.86, *P* = 0.007) and 38% reduction in the risk of heart failure (OR: 0.62, *P* < 0.001), were found in thiazide-treated patients [66]. Another recent systematic review of the Cochrane Hypertension Group of 24 randomly assigned trials with 58,040 hypertensive patients (mean age ± 62 years) shows with high-quality evidence that first-line low-dose thiazides reduced mortality (11.0% with control versus 9.8% with treatment; RR 0.89, 95% CI 0.82 to 0.97); total CVS (12.9% with control versus 9.0% with treatment; RR 0.70, 95% CI 0.64 to 0.76), stroke (6.2% with control versus 4.2% with treatment; RR 0.68, 95% CI 0.60 to 0.77), and coronary heart disease (3.9% with control versus 2.8% with treatment; RR 0.72, 95% CI 0.61 to 0.84) [67]. Additionally, thiazide and loop diuretics are important tools in the therapy of volume-overload conditions, such as congestive heart failure, nephrotic syndrome, and cirrhosis, by improving the symptoms of fluid congestion, volume overload, and edema. Although thiazide-type diuretics are among the best tolerated antihypertensive drugs they are often associated related adverse side effects, such as electrolyte, acid-base and/or metabolic disorders (e.g., impaired glucose tolerance, dyslipidemia) [68]. All types of diuretics promote excretion of sodium. Depending upon the site and mode of action, some diuretics increase excretion of potassium, magnesium, chloride, calcium, or bicarbonate (Figure 2). In general, electrolyte disorders, such as hyponatremia and hypokalemia are well considered and monitored in clinical practice therefore they are not further discussed at this point. Instead, to provide a review of current knowledge, I will focus on previously more neglected drug–nutrient interactions between thiazide-type diuretics and magnesium.

#### 4.2.1. TD and Magnesium

In general, TD are associated with a decrease of serum magnesium levels by 5% to 10%. Whereby the drug-induced magnesium depletion is be more severe in the elderly. Up to 50% of treated patients have cellular magnesium depletion, regardless of normal serum concentrations. Hypomagnesemia occurs more often in the elderly, and in those receiving continuous high-those diuretic therapy which may increase cardiovascular morbidity and mortality [69,70]. About 80% of hypertensive patients treated for at least 6 months with hydrochlorothiazide have been found to have magnesium depletion based on retention of a parenterally administered magnesium load, even though their magnesium serum levels were normal [71]. In an elderly population of a Somerset village 48% of the thiazide-treated patients were hypomagnesemic and 28% of the thiazide-treated patients were hypokalemic. Thus, magnesium and potassium depletion are commonly associated with thiazide therapy in the elderly [72,73]. Hypomagnesemia is often associated with hypokalemia, hypocalcemia, hypophosphatemia and hyponatremia [74]. Hypokalemia, hypocalcemia and/or hypovitaminosis D found in association with low serum magnesium blood levels can prove refractory to all treatment measures until the underlying magnesium deficiency is corrected [58,75]. Remarkable are the results of a cross-sectional study in hypertensive patients that determined serum and mononuclear cell magnesium concentrations. This study shows that although the patients had normal serum magnesium, TD can induce intracellular magnesium depletion not detectable by assessment of blood serum [76]. Therefore, the serum magnesium level reflects only a small part of total body content magnesium. In a patient with clinical magnesium deficiency cellular magnesium concentration can be low despite normal magnesium levels in blood serum [77].

Furthermore, it has been shown that magnesium is a kind of second messenger for insulin action. Magnesium plays a crucial role in glucose and insulin metabolism, mainly through its impact on tyrosine kinase activity of the insulin receptor, by transferring the phosphate from ATP to protein. Magnesium may also affect phosphorylase b kinase activity by releasing glucose-1-phosphate from glycogen. In addition, magnesium may directly affect glucose transporter protein activity 4 (GLUT4), and help to regulate glucose translocation into the cell [59,61,78]. Intracellular magnesium deficiency may affect the risk of insulin resistance and alter the glucose entry into the cell. It is imaginable that the subclinical magnesium deficiency and intracellular magnesium depletion associated with thiazide treatment may interfere with the activity of the tyrosine kinase and the insulin receptor increasing the risk of insulin resistance. Patients with magnesium deficiency show a more rapid progression of glucose intolerance and have an increased risk of insulin resistance. The supplementation of magnesium may contribute to an improvement in both islet Beta-cell response and insulin action in thiazides treated patients and in type-2 diabetics [59,78,79,80]. In a recent randomized, double-blind, clinical study with thiazide-treated hypertensive women (age: 40–65 years) the effects of magnesium supplementation (600 mg/day) on blood pressure and vascular function were evaluated. After 6 months, the magnesium group had a significant reduction in systolic (SBP: 144 ± 17 vs. 134 ± 14 mmHg, *P* = 0.036) and diastolic blood pressure (DBP: 88 ± 9 vs. 81 ± 8 mmHg, *P* = 0.005), and a sign of improved endothelial function a significant increase of brachial flow-mediated dilatation (FMD) (r = 0.44, *P* = 0.011). The constant oral supplementation of magnesium was associated with better blood pressure control, improved endothelial function and amelioration of subclinical atherosclerosis in these thiazide-treated hypertensive women [28,81,82,83].

#### 4.2.2. Recommendation for Clinical Practice

Subclinical magnesium deficiency is a principal driver of cardiovascular diseases such as arrhythmias, arterial calcifications, atherosclerosis, heart failure, hypertension, and/or thrombosis. In other words, disorders of magnesium metabolism are a principal, under-recognized, driver of cardiovascular disease in medical practice everyday life. Hypertensive patients on long-term treatment with TD should therefore be monitored for magnesium deficiency, particularly those with additive risk factors, such as age >60, hydrochlorothiazide doses ≥25 mg/day, insulin resistance, cardiovascular diseases (e.g., hypertension, arrhythmias), inadequate dietary intake, secondary aldosteronism, and kidney dysfunction. Dietary and supplement recommendations for magnesium please see Section 4.1.3.

## 5. Conclusions

Given the ever-increasing number of drugs on the market and the frequency with which they are used, greater attention must be paid by physicians and pharmacists in daily medical and pharmaceutical practice focused in particular on the adverse effects of drug therapy on the magnesium status in order to minimize the potential risk to the health of patients. Especially high-risk patients (e.g., children, the elderly, patients with diabetes, patients with hypertension, patients on polypharmacotherapy) and individuals under long-term medication with drugs such as PPIs or diuretics should be monitored for drug-induced magnesium deficiency. Health care providers should take the initiative to increase awareness of magnesium deficiency and encourage the general population to consume magnesium-containing foods to reduce disease burden.

## Figures and Tables

**Figure 1 ijms-20-02094-f001:**
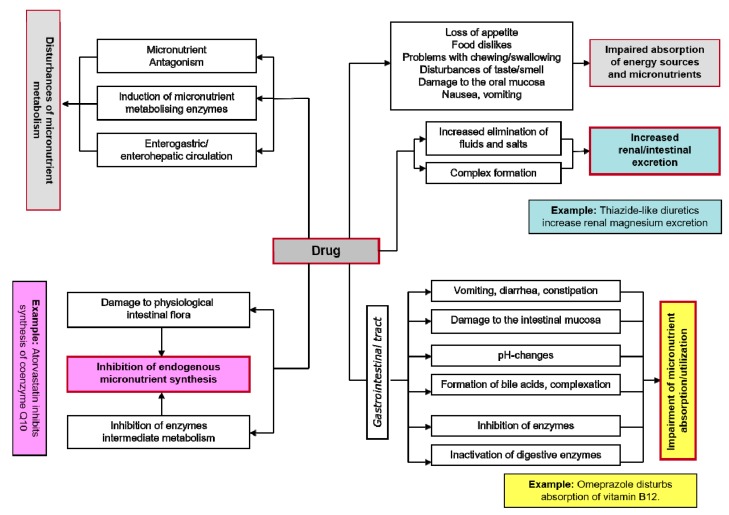
Disruption of micronutrient status by drugs [9,10].

**Figure 2 ijms-20-02094-f002:**
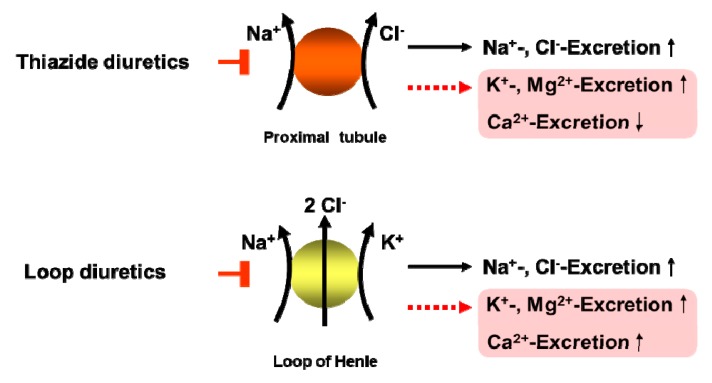
Electrolyte disorders by thiazides and loop diuretics [9,10,28].

**Table 1 ijms-20-02094-t001:** Drug-induced magnesium loss and hypomagnesemia [10,27,28,33,34,39,41,42].

Drug Group (Drug, Substance)	Examples	Mechanism/Effect
**Anti-diabetic medication**	Insulin, insulin mimetic drugs	interference with Na^+^/Mg^2+^ exchanger SLC41A1, increased renal magnesium loss
**Antimicrobials**	Aminoglycoside antibiotics (e.g., gentamicin, tobramycin, amikacin)	reduced paracellular reabsorption of magnesium; increased renal magnesium loss, secondary hyperaldosteronism
	Antimicrobial medication (e.g., Pentamidine)	increased renal magnesium loss
	Antiviral medication (e.g., foscarnet)	magnesium chelating, nephrotoxicity, increased renal magnesium loss
	Polyene antifungals(e.g., amphotericin B)	nephrotoxicity, increased renal magnesium loss
**Beta adrenergic agonists**	Fenoterol, salbutamol, theophylline	increased renal magnesium excretion, metabolic abnormalities (magnesium shift into cells)
**Bisphosphonates**	Pamidronate	renal impairment, increased magnesium excretion
**Cardiac glycoside**	Digoxin	reduced renal tubular reabsorption of magnesium, increased magnesium excretion
**Chemotherapeutic agents**	Amsacrine, cisplatin	nephrotoxicity, cisplatin accumulates in renal cortex, increased renal magnesium loss, reduced TRPM6 expression (?)
**Diuretics**	Thiazide diuretics (e.g., HCT)	reduced TRPM6 expression (distal), increased renal magnesium loss, secondary hyperaldosteronism
	Loop diuretics (e.g,. furosemide)	reduced paracellular magnesium reabsorption (thick ascending limb), increased renal magnesium loss, secondary hyperaldosteronism
**EGFR-Inhibitors**	Cetuximab	increased renal magnesium loss, reduced TRPM6 activity
**Immunosuppressants**	mTOR-Kinase-Inhibitor (e.g., Rapamycin/Sirolimus)	reduced paracellular magnesium reabsorption
	Calcineurin inhibitors (e.g., cyclosporine, tacrolimus)	reduced TRPM6 expression (distal), increased renal magnesium loss
**Proton-pump inhibitors**	Omeprazole, pantoprazole	inhibition of active magnesium absorption by interfering with TRPM6 and TRPM7, increased renal magnesium loss (?)

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
