# Peer review of "Magnesium and Drugs"

_ijms, 2019, doi:10.3390/ijms20092094_

Round 1

Reviewer 1 Report

The review is concise and informative. I do not have much comments as this review is intended more to address medical professional than scientists. Regardless the author should consider to extend the review with a paragraph describing current knowledge of interference between commonly prescribed drugs and known Mg2+ transport mechanisms and Mg homeostatic factors. Just to give an example some antipsychotic drugs and antidepressants do interfere with NME SLC41A1. Also insulin and insulomimetic drugs have potency to modulate rate and quantitative aspects of SLC41A1mediated Mg2+-efflux. Also signalome modulating drugs are known to impact on TRPM7 performance. 

Author Response

I thank the reviewer for pointing out the importance of magnesium transport mechanisms (e.g. SLC41A1). I have also included this important aspect and compiled a new table 1. I have  marked for better reading the new text in blue. Kind regards, Uwe Gröber

Reviewer 2 Report

This is an interesting manuscript focusing on the clinically important aspects of magnesium and drug interaction. With emerging drugs, magnesium wasting is increasingly becoming a clinical challenge. The author has made an effort to provide an overview of magnesium-wasting drugs. The readership would be benefitted from further clarifications of the following areas:

1.Effects of pH on magnesium-drug interactions, & how it might impact renal & intestinal magnesium uptake should be explained. 

2.Clinically, identifying hypomagnesemia is often difficult & alternate to serum measurements, should be briefly discussed (Nutrients. 2018 Dec 2;10(12). pii: E1863).

3.The safety concern related to magnesium or drug overdose should be briefly included, particularly in infants, where metabolize drugs are likely to be slower, because of the evolvement of the elementary tract.

4.The potential impact of lifestyle (smoking, alcohol, obesity) on magnesium-drug interactions would help the readership to appreciate this clinically important issue, even better.

5.Recommendations to reduce the drug-magnesium interaction, perhaps by providing time intervals between the consumption, or other feasible measures should be elaborated.

Author Response

I thank the reviewer for further clarifications on the following areas: Very important influence factors such as age, pH value and life style factors (e.g. diet, alcohol). Therefore, I included a new section 3. Influencing factors. I have  marked for better reading the new text in blue. Kind regards, Uwe Gröber

Round 2

Reviewer 2 Report

The author has adequately addressed my earlier suggestions.